# Validating Semantic Robustness of Deep Neural Networks in Colour Fundus Imaging

## Abstract

Despite the success of Deep Neural Networks (DNNs) in ophthalmic tasks, their robustness in real-world clinical settings remains uncertain. This paper presents a case study on the semantic robustness of DNN models for colour fundus imaging. We first introduce a novel optimisation algorithm, DIRECT-LSR, to identify worst-case robustness against clinically relevant semantic perturbations, including geometric transformation, illumination distortion, and motion blur. Our method provides a reliable lower bound with theoretical guarantees, enabling a practical black-box robustness validation approach. Evaluating various commonly used DNN models on colour fundus datasets, we demonstrate their vulnerability to semantic perturbations, particularly geometric transformations that drastically reduce model accuracy despite preserving clinically relevant features. As a secondary contribution, we show that a randomised data augmentation strategy can serve as an effective and accessible defence mechanism to improve models' reliability. However, since the performance gaps between clean and perturbed images persist, our results also highlight the need for more advanced defences in future work, offering insights for developing more reliable artificial intelligence systems.

## 1 Introduction

Medical diagnosis is a safety-critical task that requires Artificial Intelligence (AI) models to be both reliable and trustworthy. However, Deep Neural Network (DNN)-based AI models are known to be highly sensitive to various perturbations (Goodfellow et al., 2014; Wang et al., 2023b; Baek et al., 2024; Luo et al., 2024), raising concerns about their robustness in real-world clinical settings (Javed et al., 2025). In this work, we conduct a case study on the reliability of DNN models in Colour Fundus Photography (CFP), a cornerstone of ophthalmology for diagnosing a wide range of eye diseases (Grzybowski et al., 2024). Recent DNN models have demonstrated strong performance in CFP-based diagnosis (Grzybowski et al., 2024; Weng et al., 2024) and image quality assessment (Shen et al., 2020), yet their robustness under clinically realistic perturbations lacks comprehensive validation. The conventional approach to evaluating robustness involves formulating the generation of perturbations as an optimisation problem, thereby identifying worst-case examples to test a model's performance. (Goodfellow et al., 2014; Wang et al., 2023b). Yet this line of work faces fundamental challenges in CFP. First, algorithmic perturbations, especially pixel-level noise (Madry et al., 2018), rarely correspond to any biological or imaging artefact, limiting their clinical plausibility (Bortsova et al., 2021). Second, authentic distortions in fundus imaging arise from various factors, such as lens artefacts, eye movement, and device noise, resulting in complex perturbations that are difficult to formulate and optimise (Shi et al., 2022). Third, despite attempts to introduce semantic perturbations in medical imaging (Zhang et al., 2022b), there is still no principled framework for evaluating worst-case robustness under clinically relevant distortions. Existing approaches often rely on randomly generated natural corruptions (Hendrycks et al., 2019), which may preserve some realism but fail to capture worst-case perturbations.

To address these gaps, we propose a practical framework for evaluating the semantic robustness of DNN models in CFP tasks. As shown in Fig. 1, our approach focuses on three typical distortions in CFP images acquisition, including geometric transformations, illumination distortions, and motion blur, from an adversarial perspective. Methodologically, we introduce DIRECT-LSR, which combines DIRECT optimisation (Jones et al., 1993; Gablonsky, 2001) and the Least Squares Regression (LSR) estimation (Huang et al., 2023). Compared to first-order optimisation (Madry et al.,

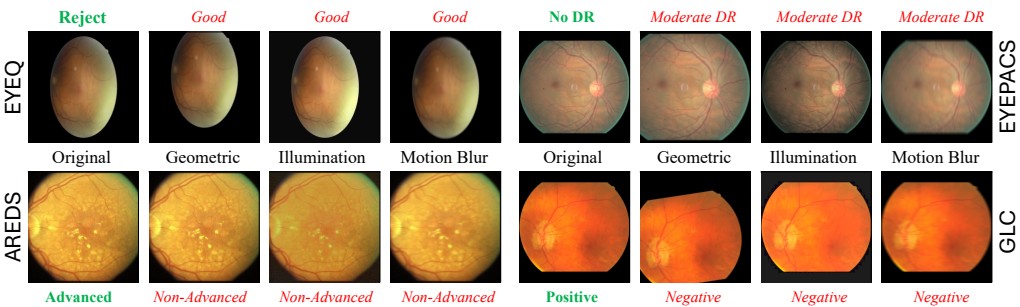

Figure 1: Semantic perturbation examples across four tasks: image quality assessment (EYEQ), DR grading (EYEPACS), and AMD/glaucoma diagnosis (AREDS/GLC). Model predictions on original images and perturbed images are shown in green and red, respectively.

2018) used for adversarial attacks, which is ineffective for optimising geometric perturbations and often performs worse than random search (Engstrom et al., 2019), DIRECT-LSR efficiently identifies near-optimal perturbations while providing provable lower-bound estimates of worst-case model performance. In contrast to formal verification approaches (Balunović et al., 2019; Mohapatra et al., 2020), DIRECT-LSR offers a practical black-box alternative that scales to diverse DNN architectures While, in theory, DIRECT-LSR could achieve sound verification given sufficiently many queries, such guarantees cannot be ensured under limited query budgets. Regarding the CFP-related tasks, we consider four common scenarios, including Diabetic Retinopathy (DR) grading (Weng et al., 2024), the diagnosis of Age-related Macular Degeneration (AMD) (Pirbhai et al., 2005) and glaucoma (Ting et al., 2017), and image quality check (Shen et al., 2020). To ensure clinical plausibility, we work with ophthalmic experts to assess the perturbed CFP images and to define semantically meaningful perturbation strengths that align with real-world conditions and human perspective.

Our experimental results show that semantic perturbations significantly reduce the accuracy of various model architectures, with geometric transformations having a more notable negative impact than illumination distortion and motion blur. Intriguingly, two ophthalmic experts, when examining the same perturbed images, found that geometric transformations had only a minor effect on their clinical validation, while severe illumination and motion blur may obscure key diagnostic features. This finding highlights the discrepancy between the models' vulnerability and human perception and enables the alignment between our algorithmic evaluation and clinical reality.

In summary, we present a case study on the semantic robustness of DNN models for colour fundus imaging. On the methodological side, we propose a novel variant of DIRECT optimisation, namely DIRECT-LSR, which incorporates local least squares regression to estimate local Lipschitz constants. Furthermore, as shown in Table 2, we find that a randomised data augmentation strategy can substantially enhance the reliability of DNN models against semantic perturbations. While the enhanced models demonstrate strong verified robustness against illumination distortions, achieving similar robustness under geometric transformation and motion blur remains challenging. This highlights the need for more effective, domain-specific defence strategies to ensure the clinical reliability of AI systems.

**Related Works**  The robustness and reliability of AI models for medical tasks have received increasing attention in recent years. While much work has focused on adversarial robustness (Goodfellow et al., 2014; Madry et al., 2018) in medical diagnosis (Javed et al., 2025) and segmentation (Cho et al., 2023; Luo et al., 2024; Wang et al., 2023a), robustness to semantic perturbations, *e.g.*, geometric transformations, illumination changes, and blur, is arguably more relevant to real-world clinical practice (Shi et al., 2022). Existing studies show that semantic corruptions naturally occurring during image acquisition can substantially degrade the performance of colour fundus imaging models in real-world settings (Shi et al., 2022; Zhang et al., 2024). However, to the best of our knowledge, there has been no systematic investigation of semantic robustness for fundus photography models.

On the methodology side, early work by Hendrycks et al. (2019) studied robustness to natural semantic corruptions using randomly generated perturbations. Originally proposed for ImageNet, this idea has since been adopted in domain-specific settings, including autonomous driving (Kong et al., 2023; Xie et al., 2023) and medical image classification (Zhang et al., 2022b). However, these ap-

proaches rely on randomly perturbed datasets and therefore cannot meaningfully assess worst-case robustness (Hendrycks et al., 2019; Zhang et al., 2022b). Achieving verified semantic robustness remains challenging: white-box verification methods (Zhang et al., 2018; Balunović et al., 2019; Mohapatra et al., 2020; Wang et al., 2021; Batten et al., 2024; Brückner & Lomuscio, 2025) struggle with semantic perturbations and scale poorly to large models, while randomised smoothing (Li et al., 2021; Hao et al., 2022) offers only probabilistic guarantees and cannot identify worst-case errors. More recently, black-box optimisation has emerged as a promising alternative, enabling the generation of pixel-level and geometric perturbations to evaluate the robustness of DNN classifiers (Wang et al., 2023b).

## 2 PRELIMINARY

This section formulates the optimisation problem for semantic perturbations and introduces the semantic perturbations.

**Problem Formulation**   Given an input image $x \in \mathbb{R}^{V \times W \times 3}$ with resolution $V \times W$ and three colour channels, along with its label $y \in \{1, \ldots, C\}$, our goal is to find the optimal semantic perturbation $P_\theta$ that changes the prediction of $x$ from a well-trained classifier $F : \mathbb{R}^{V \times W \times 3} \to \mathbb{R}^C$. The objective function $\mathcal{L}(\theta; F, x, y) : \mathbb{R}^n \to \mathbb{R}$ in semantic robustness evaluation can be written as

$$\mathcal{L}(\theta; F, x, y) = F(P_\theta(x))_y - \max_{c \in \{1, \ldots, C\} \setminus \{y\}} F(P_\theta(x))_c, \tag{1}$$

where the margin loss (Carlini et al., 2017) is adopted due to its sensitivity to the correctness of the model's prediction. Finding the optimal perturbation can then be formulated as a minimisation problem given by

$$\theta^* = \arg \min_{\theta \in \Theta} \mathcal{L}(\theta; F, x, y), \tag{2}$$

where $\Theta$ is a $n$-dimensional bounded perturbation space that contains all feasible setting $\theta$. Notably, if $\mathcal{L}(\theta^*; F, x, y) > 0$, we can conclude that the model $F$ was not fooled by perturbation $P_{\theta^*}$ on the paired image and label $(x, y)$, proving the model's robustness.

**Semantic Perturbations**   To ensure both clinical relevance and mathematical tractability, we focus on three representative perturbations that continuously challenge retinal diagnosis, *i.e.*, geometric transformations (Adal et al., 2015), illumination distortion (Mitra et al., 2018), and motion blur (Williams et al., 2017). In practice, we introduce a parameter $\gamma$ to control perturbation strength, with the bounds of each perturbation summarised in Table 1.

*Geometric transformations* are implemented via the intrinsic matrix on the pixel coordinates. For example, a pixel located at $(\mathrm{x}_j, \mathrm{y}_j)$ in the original image $x$ is mapped to $(\mathrm{x}'_i, \mathrm{y}'_i)$ in the perturbed variant $x'$ as

$$\begin{bmatrix} \mathrm{x}'_j \\ \mathrm{y}'_j \end{bmatrix} = \begin{bmatrix} \theta_s^{hor} \cdot \cos \theta_r & -\theta_s^{vrt} \cdot \sin \theta_r & \theta_t^{hor} \\ \theta_s^{hor} \cdot \sin \theta_r & \theta_s^{vrt} \cdot \cos \theta_r & \theta_t^{vrt} \end{bmatrix} [\mathrm{x}_i, \mathrm{y}_i, 1]^\top. \tag{3}$$

Here, $\theta_r$ represents the rotation angle, and we denote $\theta_s^{hor}$ and $\theta_s^{vrt}$ as the scaling factors, $\theta_t^{hor}$ and $\theta_t^{vrt}$ as the shifting factors, where the superscripts $hor$ and $vrt$ indicate the horizontal and vertical directions, respectively.

*Illumination distortion* in colour fundus imaging would be inevitable due to hardware variation and lighting conditions. We simulate the illumination changes in CFP images by adjusting the brightness and contrast. Brightness is commonly modified in the HSV colour space by shifting the brightness channel $x^{brt}$ with $\theta^{brt}$ (Mohapatra et al., 2020; Zhang et al., 2022a), which can be written as $\min(\max(x^{brt} + \theta^{brt}, 0), 1)$. The contrast adjustment is performed in the RGB colour space by scaling with $\theta^{cnt}$ (Zhang et al., 2022a), and this perturbation is given by $\min(\max(x \cdot \theta^{cnt}, 0), 1)$. In line with Retinex theory (Jobson et al., 1997), we sequentially apply brightness and contrast modifications to generate complex photometric distortions that partially mimic human visual adaptation to illumination changes.

*Motion Blur* is a common distortion in real-world CFP imaging, typically caused by relative movement between the patient's eye and the camera. We model motion blur as a convolution with a directional blur kernel (Riba et al., 2020). Specifically, the kernel is constructed by embedding a 1-D weight vector $\boldsymbol{w}$ of length $\theta^{ks}$. The elements of $\boldsymbol{w}$ are given by $\boldsymbol{w}_i = d + (1 - 2d)/(\theta^{ks} - 1) \cdot i$,

Table 1: The bounds of each perturbation factor.

| Perturbation | Hyperparameters |
|---|---|
| Geometric | $\theta_s^{vrt}, \theta_s^{hor} \in [-\gamma, \gamma], \theta_r \in [-\gamma\pi, \gamma\pi]\ \theta_t^{vrt}, \theta_t^{hor} \in [1-\gamma, 1+\gamma],$ |
| Illumination | $\theta^{brt} \in [-\gamma, \gamma],\ \theta^{cnt} \in [1-\gamma, 1+\gamma]$ |
| Motion Blur | $\theta^{ks} = \gamma,\ \theta^{ang} \in [-\pi, \pi],\ \theta^{dir} \in [-1, 1]$ |

where $d = (\theta^{dir} + 1)/2$. This vector is then embedded into the centre of a $\theta^{ks} \times \theta^{ks}$ matrix, subsequently rotated by an angle $\theta^{ang}$, and finally normalised. In practice, we fix the blur kernel size $\theta^{ks}$ and optimise its angle $\theta^{ang}$ and direction $\theta^{dir}$ to maximise impact on the target models.

## 3 VALIDATION VIA DETERMINISTIC OPTIMISATION

This section outlines the DIRECT optimisation for robustness validation and introduces a novel lower-bound estimation method based on local least squares regression.

### 3.1 DIRECT OPTIMISATION OVERVIEW

DIRECT is a gradient-free global optimisation method. The algorithm iteratively trisects the normalised search space $\Theta$ and locates Potential Optimal (PO) partitions for further exploration (Gablonsky, 2001). We denote the PO partition by $\mathbb{P}$ and the maximum divide level by $H$, *i.e.*, the maximum number of trisections allowed per dimension. We write $\sigma^i$ as a short-hand of $\sigma(\Theta^i) = \|\Theta^i\|_p$, representing an $L_p$ norm of $\Theta^i$ and let $\sigma_H$ denote the smallest partition size, where each dimension reaches the maximum divide level.

Consider a PO partition $\Theta^i$ at level $h < H$, which has $m \in \mathbb{N}_{\leq n}$ dimensions with side length $3^{-h}$ and $n - m$ dimensions with side length $3^{-h-1}$. The algorithm samples two points along each of the $m$ dimensions with longer edges, given by $\theta^i \pm 3^{-h-1}\mathbf{e}_j$, for $j \in \{1, \dots, m\}$, where $\theta^i \in \mathbb{R}^n$ is the centre of $\Theta^i$ and $\mathbf{e}_j$ is a unit vector along $j$-th dimension. These sampled points are then queried, and the DIRECT algorithm divides the partition based on the obtained results.

**Lemma 1.** *(Gablonsky, 2001) Given the index set $\mathbb{H}$ and a positive tolerance $\tau > 0$. Let $\mathcal{L}_{\min}$ denote the current best query result. Let $\mathbb{H}_1^p = \{q \in \mathbb{H} : \sigma^q < \sigma^p\}$, $\mathbb{H}_2^p = \{q \in \mathbb{H} : \sigma^q > \sigma^p\}$ and $\mathbb{H}_3^p = \{q \in \mathbb{H} : \sigma^q = \sigma^p\}$. A hyperrectangle $\Theta^p$ is said to be potentially optimal if*

$$\mathcal{L}(\theta^p) \leq \mathcal{L}(\theta^q), \forall q \in \mathbb{H}_3^p, \tag{4}$$

*and there is a $\tilde{K} > 0$ such that*

$$\max_{q \in \mathbb{H}_1^p} \frac{\mathcal{L}(\theta^p) - \mathcal{L}(\theta^q)}{\sigma^p - \sigma^q} \leq \tilde{K} \leq \min_{q \in \mathbb{H}_2^p} \frac{\mathcal{L}(\theta^q) - \mathcal{L}(\theta^p)}{\sigma^q - \sigma^p}, \tag{5}$$

*and*

$$\tau \cdot |\mathcal{L}_{\min}| \leq \mathcal{L}_{\min} - \mathcal{L}(\theta^p) + \sigma^p \min_{q \in \mathbb{H}_2^p} \frac{\mathcal{L}(\theta^q) - \mathcal{L}(\theta^p)}{\sigma^q - \sigma^p}. \tag{6}$$

As shown in Lemma 1, DIRECT employs three criteria to identify PO partitions. Eq. (4) selects partitions that yield the lowest objective value among those of equal size, ensuring that the current best-performing regions are retained. Eqs. (5) and (6) further filter the selected partitions based on their potential to yield improved solutions, balancing between exploration of unexplored regions and exploitation of promising areas. This strategy enables DIRECT to efficiently allocate sampling efforts across the search space and progressively refine the solution. We defer the pseudocode to Alg. 1 in the appendix and refer readers to Jones et al. (1993) and Gablonsky (2001).

More recently, Wang et al. (2023b) modified the DIRECT algorithm for robustness evaluation by introducing a lower-bound estimation of the objective function defined in Eq. (1). Their method computes the slope $\hat{K}$ between the centre and newly sampled points at each PO partition. The slopes along a single dimension are given by

$$\hat{K}_j^{i+} = \frac{|\mathcal{L}(\theta^i) - \mathcal{L}(\theta_j^{i+})|}{3^{-h-1}} \text{ and } \hat{K}_j^{i-} = \frac{|\mathcal{L}(\theta^i) - \mathcal{L}(\theta_j^{i-})|}{3^{-h-1}}, \tag{7}$$

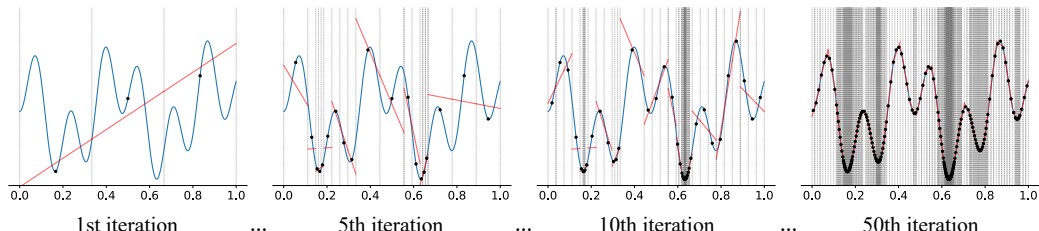

Figure 2: An illustration of DIRECT-LSR on test function: $f(z) = \frac{1}{2}(\sin(13z)\sin(27z) + 1)$, where $z \in [0, 1]$. The partitions are indicated by dashed vertical lines, and the local least squares regressions are shown as the red lines.

where $\theta_j^{i+}$ and $\theta_j^{i-}$ denote $\theta^i \pm 3^{-h-1}\mathbf{e}_j$, respectively. Then the local slope $\hat{K}^i$ of $\theta^i$ is updated as

$$\hat{K}^i = \max\{\hat{K}_{i,1}^+, \hat{K}_{i,1}^-, \ldots, \hat{K}_{i,m}^+, \hat{K}_{i,m}^-\}. \tag{8}$$

Let $\mathbb{I}$ be a set of the indices of all partitions. The lower bound of the global minimum is estimated using the found optima $\theta_o = \arg\min_{i\in\mathbb{I}}\mathcal{L}(\theta^i)$ and the largest recorded slope $\hat{K}_{\max} = \max_{i\in\mathbb{I}}\hat{K}^i$, leading to the bound as following:

$$\hat{\mathcal{L}}_{\min} = \mathcal{L}(\theta_o) - \hat{K}_{\max}\sigma_o. \tag{9}$$

## 3.2 LOCAL LEAST SQUARES REGRESSION ESTIMATION

While using the largest slope to estimate the lower bound of the objective is practical, this approach does not provide any guarantees on the tightness or soundness of the resulting bound. To address this limitation, we propose a variant of the DIRECT optimisation algorithm that incorporates Least Squares Regression (LSR) to estimate the local Lipschitz constants (Huang et al., 2023), namely DIRECT-LSR. We conduct LSR after evaluating the sampled points at each PO partition. At any PO partition $\Theta^i$ with $m$ dimensions selected for trisection, DIRECT-LSR refines the sampled points by projecting them onto the hyperplane spanned by the sampled dimensions. Specifically, each point is refined as $\tilde{\theta} = \boldsymbol{I}_m \cdot \theta$, where $\theta \in \mathbb{R}^n$ and $\boldsymbol{I}_m \in \mathbb{R}^{m \times n}$ is a projection matrix. Each row of $\boldsymbol{I}_m$ is a one-hot vector corresponding to one of the selected dimensions. Then DIRECT-LSR constructs the design matrix $\boldsymbol{X}^i \in \mathbb{R}^{(2m+1)\times(m+1)}$ as

$$\boldsymbol{X}^i = \begin{bmatrix} 1 & 1 & 1 & \cdots & 1 & 1 \\ \tilde{\theta}^i & \tilde{\theta}_1^{i+} & \tilde{\theta}_1^{i-} & \cdots & \tilde{\theta}_m^{i+} & \tilde{\theta}_m^{i-} \end{bmatrix}^\top, \tag{10}$$

and observation vector $\boldsymbol{y} \in \mathbb{R}^{2m+1}$ as

$$\boldsymbol{y}^i = \begin{bmatrix} \mathcal{L}(\theta^i) & \mathcal{L}(\theta_1^{i+}) & \mathcal{L}(\theta_1^{i-}) & \cdots & \mathcal{L}(\theta_m^{i+}) & \mathcal{L}(\theta_m^{i-}) \end{bmatrix}^\top. \tag{11}$$

Here, the dimensionality of $\boldsymbol{X}^i$ and $\boldsymbol{y}^i$ is adaptively determined by the number of dimensions being divided within the partition $\Theta^i$. Based on the design matrix and observation vector, the local LSR can be written as

$$[b^i, \beta^i] = (\boldsymbol{X}^{i\top}\boldsymbol{X}^i)^{-1}\boldsymbol{X}^{i\top}\boldsymbol{y}^i, \tag{12}$$

where $[b^i, \beta^i] \in \mathbb{R}^{d+1}$ are the regression coefficients associated with partition $\Theta^i$. Since $b^i \in \mathbb{R}$ denotes the intercept, a local Lipschitz constant can be estimated as $\hat{K}^i = \|\beta^i\|_p$, and the lower bound estimation can be written as

$$\hat{\mathcal{L}}_{\min} = \min_{i\in\mathbb{I}}(\mathcal{L}(\theta^i) - \hat{K}^i\sigma^i). \tag{13}$$

To provide intuition, we visualise the behaviour of DIRECT-LSR on a 1-D test function in Fig. 2.

## 3.3 CONVERGENCE ANALYSIS

Begin with a mild smoothness assumption, we assume the objective function $\mathcal{L}$ is differentiable and its second-order partial derivatives are upper-bounded.

**Assumption 1.** *The objective function $\mathcal{L}$ is differentiable over the $n$-dimensional perturbation space $\Theta$, and, for any $i, j \in \{1, \ldots, n\}$ and $\theta \in \Theta$, we have $\left|\frac{\partial\mathcal{L}(\theta)}{\partial\theta_i\partial\theta_j}\right| \leq \kappa$.*

Building on the sample complexity analysis from Huang et al. (2023), we can derive a bound that relates the number of samples to the gap between the estimated and real Lipschitz constant. As stated in Theorem 2, for each PO partition $\Theta^i$ with $m \in \mathbb{N}_{\leq n}$ dimensions under trisection, the LSR is performed on an $m$-dimensional hyperplane. Note that DIRECT samples and queries $2m$ points in $\Theta^i$, so, including the centre point, the total number of samples is $2m + 1$.

**Theorem 2** (Sample Complexity (Huang et al., 2023)). *Consider a $n$-dimensional PO partition $\Theta^i \in \Theta$ with $m$ dimensions selected for trisection and suppose the objective $\mathcal{L}$ satisfies Assumption 1. Then the gap between the estimated local Lipschitz constant $\hat{K}^i$ and the best Lipschitz constant $K^{i*}$ can be quantified as*

$$|\hat{K}^i - K^{i*}| \leq C \frac{\kappa \|\Theta^i\|_\infty}{\sqrt[m]{2m+1}}, \tag{14}$$

*where $C$ is a constant and $\|\cdot\|_\infty$ represents the $L_\infty$ norm.*

Theorem 2 provides a preliminary bound on this estimation gap that can be further simplified. As described in Corollary 1, $\|\Theta^i\|_\infty$ can be replaced by $3^{-h}$ according to the space trisection process of DIRECT. Because $3 \geq \sqrt[m]{2m+1} \geq 1$ holds for $m \geq 1$, we have $C \frac{\kappa 3^{-h}}{\sqrt[m]{2m+1}} \leq C\kappa 3^{-h}$. Besides, following the practice in Huang et al. (2023), we let the constant $C = (20m^{\frac{1}{p}-\frac{1}{2}})^{-1}$ in Eqs. (14) and (15), where $p$ denotes the $L_p$ norm used to compute the partition size $\sigma$.

**Corollary 1.** *Consider a $n$-dimensional PO partition $\Theta^i \in \Theta$ at level $h$ with $m$ dimensions selected for trisection and suppose the objective $\mathcal{L}$ satisfies Assumption 1. The gap between the estimated local Lipschitz constant $\hat{K}^i$ and the best Lipschitz constant $K^{i*}$ can be quantified as*

$$|\hat{K}^i - K^{i*}| \leq C\kappa 3^{-h}. \tag{15}$$

According to Corollary 1, the estimated Lipschitz constant converges exponentially to the true Lipschitz constant with respect to the divide level, *i.e.*, $\lim_{h \to \infty} C\kappa 3^{-h} = 0$. While we introduce $H$ to control the search granularity, setting $H = 6$ already yields an error bound on the order of $10^{-3}$.

## 4 EMPIRICAL STUDIES

Our experiments include three parts: (1) Evaluating the effectiveness of DIRECT-LSR in estimating reliable lower bounds; (2) Assessing the impact of semantic perturbations across varying strengths, including comparisons with baseline methods and expert validation on perturbed images; (3) Benchmarking the semantic robustness of DNNs on CFP tasks, with additional comparison on randomised data augmentation for robustness improvement.

### 4.1 GENERAL SETUP

**Datasets** For CFP-related tasks, we consider one quality assessment task and three representative diagnostic tasks in common ophthalmic scenarios that rely on CFP as the primary imaging reference: (1) Image quality assessment on **EYEQ** dataset (Shen et al., 2020) with $C = 3$ classes (Reject, Usable, Good); (2) Diabetic retinopathy (DR) grading on **EYEPACS** dataset (Dugas et al., 2015) with $C = 5$ levels (No, Mild, Moderate, Severe, Proliferative); (3) Age-related macular degeneration (AMD) classification ($C = 2$) using the first and last visit of participants in **AREDS** study (Group et al., 1999), identifying AMD severity into non-advanced and advanced AMD (Davis et al., 2005; Bridge et al., 2021); (4) Glaucoma classification ($C = 2$) using a combined dataset **GLC** based on five clinical studies (Islam et al., 2021; Zhang et al., 2010; De Vente et al., 2023; Bajwa et al., 2020; Orlando et al., 2020). The details of each datasets are summarised in the appendix (Table 3).

**DNN models** We evaluate four model architectures as targets: three convolutional backbones, *i.e.*, ResNet50 (He et al., 2016), EfficientNet (Tan & Le, 2019), and DenseNet (Huang et al., 2017), and a pure Transformer model (RetFound (Zhou et al., 2023)). On each CFP dataset, we fine-tune the target models for 80 epochs using the Adam optimiser Kingma (2014) with a learning rate of 0.0005. For RetFound, we only train the final layer on each dataset, whereas we fine-tune the entire convolutional models. During fine-tuning, the checkpoints achieving the highest test-set F1 score are selected, and their performance on the clean datasets is reported in the appendix (Table 4).

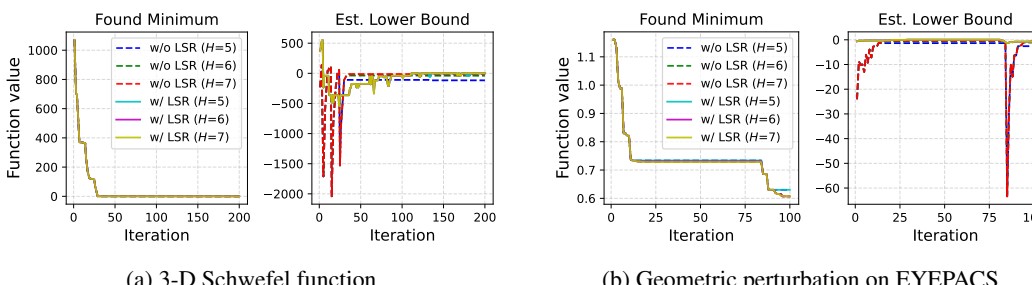

(a) 3-D Schwefel function        (b) Geometric perturbation on EYEPACS

Figure 3: Visualising the lower bound estimates derived from the proposed Least Squares Regression (LSR) method. Evaluations are performed on two tasks: (a) minimisation of the 3D Schwefel function, and (b) optimisation of geometric perturbations on the first image from the EYEPACS test set, targeting the RetFound model.

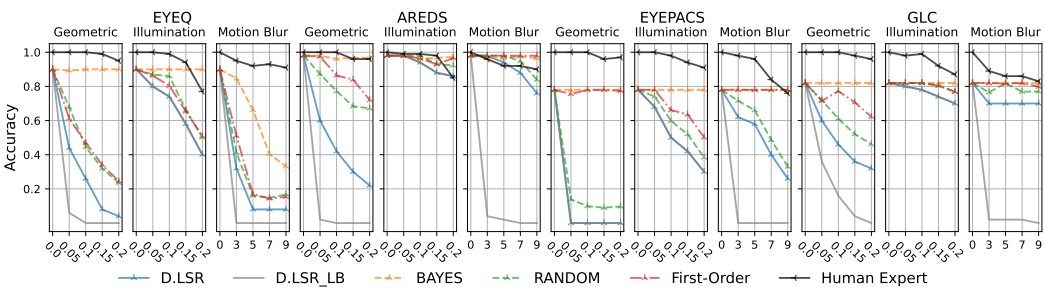

Figure 4: Accuracy of ResNet50 on perturbed CFP images, with distortions optimised by DIRECT-LSR, first-order gradient descent, Bayesian, and random search. The dark line indicates expert validation of preserved clinical features.

## 4.2 ESTIMATED LOWER BOUND AND DIVIDE LEVEL

In this experiment, we compare the lower bounds estimated by DIRECT-LSR (Eq. (13)) with those obtained using the approach of Wang et al. (2023b) (Eq. (9)). The comparison is conducted on two tasks, which are the minimisation of the 3D Schwefel function with a known global minimum of 0 and the optimisation of geometric perturbations on an EYEPACS test image against the RetFound model. As illustrated in Fig. 3, while both methods eventually reach similar minima, the LSR-based estimates are consistently tighter and more stable across both tasks. We also compare the results under different maximum levels $H \in 5, 6, 7$. As shown in Fig. 3b, the found minima at $H = 5$ are consistently worse than those obtained at $H = 6$ and $H = 7$, indicating insufficient search granularity. In contrast, the results for $H = 6$ and $H = 7$ are nearly identical, suggesting that increasing the level beyond 6 yields diminishing returns. Therefore, we set $H = 6$ as the default, offering a reliable and efficient trade-off for subsequent evaluations.

## 4.3 PERTURBATION IMPACT AND BASELINE COMPARISON

In this experiment, we compare the DIRECT-LSR to baseline methods and evaluate the target models' semantic robustness at different perturbation strengths. The geometric and illumination perturbations are carried out at $\gamma \in \{0.05, 0.1, 0.15, 0.2\}$. The kernel size of motion blur is set to $\gamma \in \{3, 5, 7, 9\}$. We conduct experiments on 50 uniformly sampled images from each of the four CFP tasks, applying three types of semantic perturbations at varying strengths. The target model here is the ResNet50. We perform DIRECT-LSR optimisation with up to 2000 queries, using $H = 6$. The average runtime is approximately 4.2 seconds per example. For comparison, we evaluate three additional baselines: Bayesian optimisation, random search, and a first-order adversarial attack. Bayesian optimisation uses the expected improvement acquisition function (Frazier, 2018) and requires additional runtime to update the Gaussian process surrogate model, averaging 95.7 seconds per example. Random search is performed with 2000 queries, returning the best result among the

Table 2: Semantic robustness benchmark on CFP tasks. All models share the same training setup; models marked "+ Aug." additionally apply random semantic perturbation as data augmentation. Best results are in **bold**.

| Model | Metric | EYEQ | | | | EYEPACS | | | | AREDS | | | | GLC | | | |
|---|---|---|---|---|---|---|---|---|---|---|---|---|---|---|---|---|---|
| | | Clean | Geo. | Illu. | MB | Clean | Geo. | Illu. | MB | Clean | Geo. | Illu. | MB | Clean | Geo. | Illu. | MB |
| Retfound | Acc. | 0.896 | 0.148 | 0.488 | 0.18 | 0.808 | 0.438 | 0.71 | 0.622 | 0.948 | 0.184 | 0.842 | 0.844 | 0.894 | 0.42 | 0.616 | 0.63 |
| | $\hat{\mathcal{L}}_{\min} > 0$ | / | 0.108 | 0.48 | 0.002 | / | 0.042 | 0.71 | 0.004 | / | 0.114 | 0.842 | 0.064 | / | 0.18 | 0.616 | 0.004 |
| RetFound + Aug. | Acc. | 0.872 | **0.596** | 0.696 | 0.788 | 0.802 | **0.56** | **0.742** | 0.748 | 0.94 | 0.662 | 0.872 | 0.874 | 0.886 | 0.488 | 0.68 | 0.792 |
| | $\hat{\mathcal{L}}_{\min} > 0$ | / | **0.386** | 0.696 | 0.064 | / | **0.3** | **0.742** | **0.102** | / | 0.292 | 0.872 | 0.017 | / | **0.28** | 0.68 | 0.072 |
| ResNet50 | Acc. | **0.912** | 0.116 | 0.51 | 0.156 | 0.77 | 0.0 | 0.39 | 0.398 | **0.952** | 0.258 | 0.898 | 0.88 | 0.928 | 0.218 | 0.866 | 0.82 |
| | $\hat{\mathcal{L}}_{\min} > 0$ | / | 0.054 | 0.508 | 0.034 | / | 0.0 | 0.386 | 0.0 | / | 0.0 | 0.898 | 0.006 | / | 0.004 | 0.86 | 0.006 |
| ResNet50 +Aug. | Acc. | 0.898 | 0.428 | 0.744 | 0.83 | 0.838 | 0.356 | 0.646 | 0.758 | 0.936 | 0.658 | 0.906 | 0.908 | 0.956 | 0.476 | **0.902** | **0.89** |
| | $\hat{\mathcal{L}}_{\min} > 0$ | / | 0.208 | 0.744 | 0.112 | / | 0.092 | 0.644 | 0.028 | / | 0.184 | 0.902 | 0.16 | / | 0.028 | **0.9** | 0.024 |
| EfficientNet | Acc. | 0.896 | 0.136 | 0.724 | 0.164 | 0.786 | 0.0 | 0.24 | 0.416 | 0.938 | 0.238 | 0.896 | 0.806 | 0.936 | 0.264 | 0.868 | 0.676 |
| | $\hat{\mathcal{L}}_{\min} > 0$ | / | 0.024 | 0.724 | 0.014 | / | 0.0 | 0.238 | 0.0 | / | 0.0 | 0.896 | 0.002 | / | 0.002 | 0.868 | 0.002 |
| EfficientNet + Aug. | Acc. | 0.882 | 0.518 | 0.534 | 0.77 | **0.852** | 0.43 | 0.596 | **0.768** | 0.942 | **0.698** | 0.92 | **0.926** | 0.946 | **0.554** | 0.894 | 0.88 |
| | $\hat{\mathcal{L}}_{\min} > 0$ | / | 0.228 | 0.534 | 0.084 | / | 0.134 | 0.584 | 0.034 | / | 0.13 | **0.92** | 0.226 | / | 0.176 | 0.894 | 0.052 |
| DenseNet | Acc. | 0.9 | 0.136 | **0.73** | 0.16 | 0.784 | 0.01 | 0.632 | 0.572 | 0.948 | 0.406 | 0.896 | 0.876 | 0.94 | 0.188 | 0.86 | 0.788 |
| | $\hat{\mathcal{L}}_{\min} > 0$ | / | 0.054 | **0.73** | 0.046 | / | 0.0 | 0.632 | 0.0 | / | 0.086 | 0.896 | 0.07 | / | 0.0 | 0.86 | 0.002 |
| DenseNet + Aug. | Acc. | 0.898 | 0.422 | 0.702 | **0.86** | 0.828 | 0.31 | 0.718 | 0.758 | 0.944 | 0.674 | 0.908 | 0.914 | **0.964** | 0.45 | 0.9 | 0.886 |
| | $\hat{\mathcal{L}}_{\min} > 0$ | / | 0.242 | 0.702 | **0.128** | / | 0.066 | 0.716 | 0.036 | / | **0.414** | 0.908 | **0.456** | / | 0.114 | 0.9 | **0.082** |

sampled perturbations. Its average runtime is 4 seconds per example. Finally, we implement a first-order adversarial attack similar to projected gradient descent (Madry et al., 2018), using 50 iterations with a step size equal to one tenth of the available perturbation range, resulting in an average runtime of 0.35 seconds per example. While the first-order attack is faster than DIRECT-LSR and random search in terms of runtime, it requires access to the models' parameters and GPUs for backpropagation. In alignment with clinical practice, two ophthalmic experts evaluate whether the perturbed CFP images preserved essential features required for diagnosis. We provide an illustration of the marking interface in the appendix. The results are summarised in Fig. 4.

**DIRECT-LSR vs. baselines** We can observe from Fig. 4 that DIRECT-LSR consistently identifies stronger semantic perturbations than baseline methods, while also exhibiting greater stability across tasks. Random search performs competitively at lower distortion levels but exhibits high variability. First-order adversarial attacks achieve strong results in certain tasks but remain unstable across datasets and perturbation types. Due to the lack of a convergence guarantee, both random search and first-order attack may achieve worse performance as $\gamma$ increases. Bayesian optimisation, on the other hand, often fails to locate effective perturbations, highlighting its limitation in optimising the semantic perturbation. Notably, we find that the first-order attack, commonly used to evaluate robustness against pixel-level perturbations (Madry et al., 2018), consistently performs worse than simple random search. This observation is consistent with Engstrom et al. (2019), which reported that random search can outperform first-order attacks under geometric perturbations. Our results further show that this phenomenon also occurs under illumination and motion-blur perturbations.

**Perturbation Effectiveness** All perturbations show a strong negative impact on the target models' accuracy as $\gamma$ increases. Geometric perturbations cause the most significant drop in performance across all tasks. It drops the model's accuracy to zero on the EYEPACS dataset. Illumination perturbations notably affect the models' performance on EYEQ, EYEPACS, and GLC. Motion blur leads to relatively mild performance degradation on AREDS, EYEPACS, and GLC, and shows a comparable impact to geometric perturbations on EYEQ, highlighting the task and model-specific sensitivities. In addition, the grey solid line (D.LSR_LB) indicates the number of examples with a positive estimated lower bound. As the perturbation strength increases, this line consistently drops across datasets and distortion types, reflecting the growing difficulty of maintaining robustness guarantees under stronger perturbations. Compared to accuracy curves, the grey line provides a complementary view of model reliability, highlighting cases where robustness cannot be certified even if empirical accuracy remains moderate, especially with the geometric perturbation and motion blur.

**Expert Validation** To align with clinical practice, we invite two ophthalmic experts to assess the perturbed CPF images. Each expert reviews the perturbed images and determines whether key

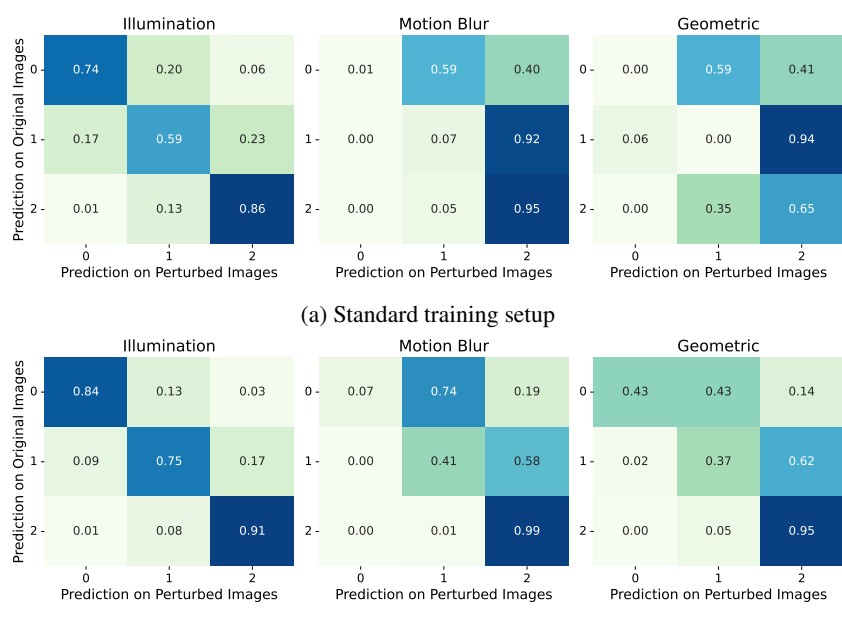

(a) Standard training setup

(b) Semantic augmentation training setup

Figure 5: Averaged prediction transition matrices on the EYEQ (image quality check) dataset comparing (a) standard and (b) augmented training strategies. The matrices show how models' predictions change under illumination, motion blur, and geometric perturbations. The labels 0, 1, and 2 correspond to good, usable, and rejected image quality, respectively. All values are row normalised.

diagnostic features are sufficiently preserved to support the original clinical label. As illustrated by the dark solid lines in Fig. 4, notable degradations are observed only under strong illumination distortions and motion blur. In contrast, even at maximum strength, geometric perturbation causes only marginal reductions in expert agreement, with illumination and motion blur at medium strength producing similarly minor effects.

## 4.4 BENCHMARK SEMANTIC ROBUSTNESS

In this experiment, we evaluate the robustness of CFP models. For testing, 500 samples are uniformly selected from the test set of each dataset. Based on expert assessments, the perturbation strengths are set to 0.2 for geometric transformations, 0.1 for illumination changes, and 5 for motion blur. While the finetuning setup remains unchanged, we introduce a simple data augmentation strategy that randomly applies valid semantic perturbations to training examples to examine whether semantic robustness can be improved (detailed in the Appendix). As reported in Table 2, we report both model accuracy and the proportion of examples with a positive lower bound.

We can see that all models exhibit substantial gains in robustness when trained with data augmentation. While the enhanced RetFound model exhibits the most noticeable robustness improvement on EYEPACS, convolution models actually achieve comparable and even better performance on other datasets. The superior performance of convolution models aligns with the finding of Zhu et al. (2024), which suggests that convolutional neural networks can outperform Vision Transformers on CFP datasets. However, since performance gaps remain, particularly under geometric perturbations, semantic robustness continues to pose a significant challenge, indicating that simple data augmentation alone is insufficient to resolve the problem.

On the other hand, robustness under illumination perturbations is easier to validate, with most correctly classified examples in both original and augmented models achieving a positive lower bound. However, for geometric and motion blur perturbations, only a few examples obtain a positive lower bound. This may be attributed to the fact that illumination distortions only introduce uniform pixel-level changes, and the inherent variability in lighting conditions within CFP encourages models to learn illumination-invariant features. In contrast, the tested models appear to be more sensitive to

geometric transformations and motion blur, even under clinically meaningful perturbation settings, making robustness validation more challenging. The vulnerability to geometric transformations may be attributed to insufficient variability in spatial configurations within the training data, whereas motion blur may obscure critical diagnostic features essential for accurate prediction.

Additionally, we conduct a case study on the EYEQ dataset for the image-quality assessment task, while the remaining analyses are deferred to the appendix. Fig. 5 visualises the averaged prediction transition caused by different semantic perturbations across all models. Among three perturbations, the illumination perturbation produces the mildest changes, indicating that global brightness fluctuations do not strongly affect the learned features. Motion blur, on the other hand, severely affects the standard trained models, where 92% of usable images are downgraded to rejected. Because our previous expert assessment shows that the motion blur at this level would not compromise clinical usability, this phenomenon suggests the standard trained models may have overfit to imaging sharpness and learnt to rely on blur as a rejection cue. For Geometric transformations, the standard trained models exhibit the problematic flip, where 35% of rejected images are classified as usable. This counterintuitive shift shows that the standard trained model may rely on unstable features that are easily disturbed by geometric perturbations, making the perturbed images' quality even appear "better." As shown in Fig. 5b, semantic augmentation generally improves the trained models' robustness. While the augmented models still show some sensitivity to motion blur, where many "good" images are considered as "usable", they are more stable under most settings, showing stronger diagonal patterns in the transition matrices. Compared with the standard trained models, they produce fewer inconsistent label flips, demonstrating the practical benefit of semantic augmentation for training clinically reliable models.

## 5 CONCLUSION AND DISCUSSION

**Summary.** This work presents a principled and clinically informed study on the semantic robustness of AI models in colour fundus imaging. We propose an optimisation-based framework for worst-case robustness evaluation under clinically relevant perturbations. Centred on the DIRECT-LSR solver, our method can achieve a tighter and more stable lower bound estimation with theoretical guarantees than existing approaches. To ensure clinical relevance, ophthalmic experts were engaged to assess perturbed images and guide the choice of meaningful perturbation strengths. Experiments across four colour fundus imaging tasks reveal that semantic perturbations can substantially degrade performance, even for domain-specific foundation models like RetFound. While a randomised data augmentation strategy effectively enhances robustness without sacrificing accuracy on clean data, semantic robustness remains an open challenge that requires further systematic investigation and more advanced defence methods.

**Limitations.** While we introduce three optimisable semantic perturbations, certain real-world distortions (e.g., lens artefacts) are not included, as they remain difficult to model and optimise. In addition, although DIRECT-LSR comes with a convergence guarantee in theory, it may fail to achieve sound verification in challenging cases, especially under limited query budget. Furthermore, this work primarily focuses on evaluating semantic robustness rather than developing new defence strategies, which we leave for future exploration. Overall, our study provides valuable insights and points toward promising research directions for reliable AI in ophthalmology, particularly in designing more realistic perturbations and reducing the performance gap between clean and perturbed inputs.

## 6 ETHICS STATEMENT

This work complies with the ICLR Code of Ethics. No human subjects or animal experiments were involved. All datasets were used in accordance with their respective usage guidelines, ensuring that privacy was not violated. No personally identifiable information was accessed, and no experiments were conducted that could raise privacy or security concerns. We have also taken care to minimise potential biases or discriminatory outcomes in the research. We remain committed to transparency, integrity, and ethical responsibility throughout the research process.

## 7 REPRODUCIBILITY STATEMENT

We confirm that all results reported in this paper are reproducible. The experimental setup—including training procedures, model configurations, and hardware specifications—is described in detail in the main paper and appendix. Upon acceptance, we will release the full source code to facilitate replication. In addition, the datasets used in this work (EYEPACS, AREDS, EYEQ, and GLC) are publicly available, enabling consistent and reproducible evaluation.

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
