## A    LLM USAGE

Large Language Models (LLMs) were used solely to support the writing and polishing of this manuscript. Specifically, we employed an LLM to refine language, improve readability, and enhance clarity in selected sections. The assistance included sentence rephrasing, grammar checking, and improving textual flow.

Importantly, the LLM was not involved in the ideation, research methodology, experimental design, or data analysis. All research concepts, methods, and results were conceived, executed, and validated entirely by the authors. The LLM's contribution was limited to improving the linguistic quality of the manuscript, with no influence on its scientific content.

The authors take full responsibility for the entire content of the paper, including any text revised with LLM assistance. We confirm that the use of LLMs complied with ethical guidelines and did not result in plagiarism or scientific misconduct.

## B    PSEUDOCODE

As a complement to Alg. 1, we detail the `PartitionSelection()` procedure in Alg. 2.

---

**Algorithm 1:** DIRECT with lower bound estimation

---

**Input:** The objective function $\mathcal{L}$, the search space $\Theta$, the number of iterations $T$, the maximum depth $H$

**Output:** The optimal factor $\tilde{\theta}$

1 Let $\mathbb{P} = \{\Theta\}$, and $t \leftarrow 0$
2 **while** $t < T$ *and* $\mathbb{P} \neq \varnothing$ **do**
3      Initialise $\mathcal{X} = \{\}$
4      **for** *each potential optimal partition $\Theta^i$ in $\mathbb{P}$* **do**
5          **if** $\sigma^i > \sigma_H$ **then**
6              **for** *dimension $j$ with long edge of $\Theta^i$* **do**
7                  Append$(\mathcal{X}, \theta^i \pm 3^{-h-1}\mathbf{e}_j)$
8      $\mathcal{Y} = \mathcal{L}(\mathcal{X})$
9      **if** $\min \mathcal{Y} < \mathcal{L}(\tilde{\theta})$ **then**
10          $\tilde{\theta} = \arg\min_{\mathcal{X}} \mathcal{Y}$
11      $\mathcal{L}_{\min}^* = $ LowerBoundEstimation()
12      **for** *each potential optimal partition $\Theta^i$ in $\mathbb{P}$* **do**
13          Trisect $\Theta^i$ based on query results in $\mathcal{Y}$
14      $\mathbb{P} = $ PartitionSelection()
15      $t = t + 1$

---

**Algorithm 2:** PartitionSelection() in DIRECT

---

**Input:** $\mathcal{L}_{\min}$ the obtained best query results, $H$ the maximum depth

**Output:** $\mathcal{P}$ a set of PO subspaces

1 Set $\mathcal{P} = \{\}$
2 **for** *each space size $\sigma > \sigma_H$* **do**
3      Select subspace $\Theta^p$ that satisfies Eqs. (4) to (6)
4      Append$(\mathcal{P}, \Theta^p)$
5 **return** $\mathcal{P}$

---

**Complexity**    The computational complexity of a single run of LSR for $2m + 1$ samples in an $n$-dimensional space is typically $\mathcal{O}(n^2 m)$ (Huang et al., 2023). In DIRECT-LSR, the number of samples used for LSR within each partition is bounded by the dimensionality, *i.e.*, $m \leq n$. Besides,

Table 3: Datasets details for the four ophthalmic tasks.

| Task | Train Size | Test Size |
|------|------------|-----------|
| Image quality check (EYEQ) | 12545 | 16251 |
| Diabetic retinopathy grading (EYEPACS) | 35125 | 53574 |
| Age-related macular regeneration (AREDS) | 8420 | 8613 |
| Glaucoma (GLC) | 4822 | 4822 |

Figure 6: An example of interface presented to ophthalmic experts

LSR is performed on all PO partitions at each iteration. Given the number of selected PO partitions grows linearly with the maximum level $H$, the overall computational complexity of the LSR is $\mathcal{O}(Hn^3)$ at each iteration.

## C  DISEASE BACKGROUND

The progression of diabetic retinopathy (DR) is related to vasculature abnormalities, including microaneurysms, hard exudates, new vessels, fibrous proliferations, and macular edema (Stitt et al., 2016). The five-stage grading system (non-DR, mild DR, moderate DR, severe DR, or proliferative DR) (Wilkinson et al., 2003) is widely adopted for the DR severity assessment.

Age-related macular degeneration (AMD) is a disease that affects the macular region of the retina, causing progressive loss of central vision (Mitchell et al., 2018). Early-stage AMD includes findings such as drusen and abnormalities of the retinal pigment epithelium, advanced AMD is defined by the presence of signs indicating either neovascular or atrophic AMD (Ferris III et al., 2013). We identify AMD severity into non-advanced and advanced AMD according to 9-steps AMD Severity Scale (Davis et al., 2005) (non-advanced AMD: scale 1-8, advanced AMD: scale 9 and with Neovascular/Central Geographic Atrophy Findings) similar to (Bridge et al., 2021).

Glaucoma is a progressive optic neuropathy characterised by the degeneration of retinal ganglion cells, manifesting the structural damage as neuroretinal rim loss, excavation, and enlargement of the optic cup in fundus images (Weinreb et al., 2014). The evaluation of glaucoma involves assessing both structural damage (cup-to-disc ratio) and functional damage (visual field loss)(Foster et al., 2002). To preserve clinically relevant features, our geometric perturbations on the GLC dataset are limited to rotation and translation, avoiding deformation of the cup-to-disc ratio.

## D  IMPLEMENTATION DETAILS

**Hardware**  Both training and evaluation are performed on a workstation equipped with an RTX 3090 graphic card, an Intel Core i9-12900K processor, and 64 GB of memory.

**Numerical Examples**  In Fig. 2, we visualise the DIRECT-LSR on a test function: $f(z) = \frac{1}{2}(\sin(13z)\sin(27z) + 1)$, where $z \in [0, 1]$. The algorithm is executed for 50 iterations with a maximum partition depth of $H = 6$. In Fig. 3a, we conduct both DIRECT and DIRECT-LSR for 200 iterations on a 3-D Schwefel function. It can be written as $f(z) = 418.9829d - \sum_{i=1}^{d} z_i \sin(\sqrt{\|z_i\|})$, where we set $d = 3$ in this 3-D case and $z \in [-500, 500]$.

---

**Algorithm 3:** Standard data preprocessing

---

**Input:** Image $x$
**Output:** Transformed image $x'$

1   $x = \text{resize}(x, (224, 224))$;
2   $x = \text{random\_horizontal\_flip}(x)$;
3   $x' = \text{normalise}(x, \text{mean} = [0.485, 0.456, 0.406],$
4                 $\text{std} = [0.229, 0.224, 0.225])$;

---

---

**Algorithm 4:** Semantic Augmentation

---

**Input:** Image $x$
**Output:** Transformed image $x'$

1   $x = \text{resize}(x, (224, 224))$;
2   $x = \text{random\_horizontal\_flip}(x)$;
3   $x = \text{random\_affine}(x, \text{degrees} = (-\frac{\pi}{5}, \frac{\pi}{5}), \text{translate} = (0.2, 0.2), \text{scale} = (0.8, 1.2))$;
4   $x = \text{color\_jitter}(x, \text{brightness} = 0.2)$;
5   **if** *random*$() < 0.8$ **then**
6     |   $x = \text{random\_motion\_blur}(x, \text{kernel\_size} = (3, 9),$
7     |       $\text{angle} = (-180, 180), \text{direction} = (-1, 1))$;
8   $x' = \text{normalise}(x, \text{mean} = [0.485, 0.456, 0.406],$
9                 $\text{std} = [0.229, 0.224, 0.225])$;
10   **return** $x'$;

---

**Sampled Test Sets**   Subsampling is commonly used for managing computational cost in robustness studies (Wang et al., 2023b). In Fig. 4, we uniformly sample 50 examples from each dataset to illustrate the performance degradation caused by semantic perturbations at varying strengths. For a more comprehensive analysis Table 2 reports results on 500 uniformly sampled examples from each dataset to assess the semantic robustness of the trained models. The dataset statistics are summarised in Table 3.

**Perturbation Effect Validation**   To enable expert validation, we reconstruct the perturbed CFP images using the optimal semantic perturbations found by DIRECT-LSR in Figure 4. As illustrated in Figure 6, we provide an interface to support ophthalmic experts in reviewing and assessing the clinical plausibility of the generated perturbations. Assuming the ground truth labels are correct, we display the original label to the experts while omitting explicit information about perturbation strength. Instead, we use indices from 0 to 3 to denote increasing levels of perturbation severity.

**Data Augmentation**   We implemented a simple data augmentation strategy using randomised image transformations from the `torchvision` toolbox. The standard data preprocessing and augmentation pipelines are detailed in Alg. 3 and Alg. 4, respectively.

Let $F$ be the model parametrised by $\mathbf{w}$, enhancing the model's semantic robustness via data augmentation can be formulated as the following minimisation problem:

$$\min_{\mathbf{w} \in \mathcal{W}} \mathbb{E}_{x \in X} \mathbb{E}_{\theta \in \Theta} \mathcal{L}_{\text{ce}}(\theta; F, x, y), \tag{16}$$

where $\mathcal{L}_{\text{ce}}$ denotes the cross-entropy loss, $\mathcal{W}$ is parameter space, and $X$ is training dataset. In Eq. (16), $\Theta$ represents the perturbation space, which includes all perturbation factors summarised in Table 1. These perturbations are randomly applied to training samples during preprocessing, before input into the DNN models. A pseudocode implementation of the proposed augmentation is provided in the Appendix. The augmentation process in Alg. 4 aligns with the training objective defined in Eq. (16), by applying geometric transformation via `random_affine`, brightness adjustments via `color_jitter`, and motion blur using a custom function with a randomised kernel size.

**Performance on Original Datasets**   In this subsection, we report the performance of the models on the original, unperturbed datasets. As shown in Table 4, all model architectures achieve high

Table 4: The model performance on clean CFP datasets.

| Model | EYEQ | | EYEPACS | | AREDS | | GLC | |
|---|---|---|---|---|---|---|---|---|
| | Acc. | F1 | Acc. | F1 | Acc. | F1 | Acc. | F1 |
| RetFound | 0.882 | 0.86 | 0.783 | 0.411 | 0.921 | 0.889 | 0.828 | 0.818 |
| +Aug. | 0.882 | 0.861 | 0.783 | 0.41 | 0.916 | 0.882 | 0.822 | 0.813 |
| ResNet50 | 0.885 | 0.864 | 0.769 | 0.453 | 0.932 | 0.904 | 0.865 | 0.856 |
| +Aug. | 0.875 | 0.855 | 0.77 | 0.452 | 0.933 | 0.905 | 0.883 | 0.876 |
| RegNet | 0.829 | 0.796 | 0.732 | 0.335 | 0.876 | 0.828 | 0.757 | 0.746 |
| +Aug. | 0.82 | 0.79 | 0.741 | 0.335 | 0.898 | 0.857 | 0.7684 | 0.757 |
| DenseNet | 0.891 | 0.872 | 0.774 | 0.474 | 0.933 | 0.908 | 0.867 | 0.861 |
| +Aug. | 0.878 | 0.858 | 0.814 | 0.475 | 0.934 | 0.906 | 0.881 | 0.875 |
| EfficientNet | 0.88 | 0.863 | 0.764 | 0.48 | 0.937 | 0.912 | 0.871 | 0.864 |
| +Aug. | 0.877 | 0.855 | 0.83 | 0.492 | 0.94 | 0.915 | 0.887 | 0.881 |
| CoAtNet | 0.863 | 0.843 | 0.759 | 0.42 | 0.922 | 0.888 | 0.814 | 0.807 |
| +Aug. | 0.861 | 0.836 | 0.766 | 0.391 | 0.925 | 0.894 | 0.81 | 0.802 |

Table 5: Comparison of DIRECT-LSR and related methods on ImageNet under geometric perturbations.

| Model | Acc. | DIRECT-$L_\infty$ | | | SimpleDIRECT | | | DIRECT-LSR | | |
|---|---|---|---|---|---|---|---|---|---|---|
| | | ASR | V. Acc. | Time(s) | ASR | V. Acc. | Time(s) | ASR | V. Acc. | Time(s) |
| ResNet50 | 78.40% | 39.54% | 40.20% | 5.0±0.5 | 40.05% | 41.80% | 4.0±1.4 | **41.33%** | 21.00% | 5.3±0.1 |
| WideResNet50 | 81.60% | 38.24% | 40.20% | 6.0±0.6 | 39.95% | 38.20% | 5.5±1.2 | **41.95%** | 17.20% | 7.4±0.6 |
| Vit$_{16\times16}$ | 81.40% | 47.91% | 36.00% | 4.9±0.8 | 49.63% | 33.60% | 3.9±0.9 | **35.31%** | 19.40% | 6.5±0.1 |
| Large Beit$_{16\times16}$ | 85.60% | 22.90% | 59.80% | 9.2±1.2 | 23.60% | 58.20% | 8.1±2.0 | **24.12%** | 46.20% | 13.4±0.2 |
| SwinTransformer | 80.20% | 55.11% | 13.00% | 5.8±0.5 | 57.61% | 14.00% | 5.4±0.9 | **59.20%** | 11.80% | 7.8±0.1 |

accuracy and F1 scores on the EYEQ, AREDS, and GLC datasets. In contrast, their performance on the EYEPACS dataset is noticeably lower, which may be attributed to the dataset's inherent class imbalance and greater visual heterogeneity. Moreover, the augmentation method in Alg. 4 helps maintain or even improve model performance on clean data.

# E   ADDITIONAL EXPERIMENTS

## E.1   MORE COMPARISON WITH DIRECT-BASED METHODS

We compare the performance of DIRECT-LSR to other DIRECT methods under the same setting in Wang et al. (2023b), which evaluated the robustness of ImageNet classifiers against geometric transformation attacks. Different to our setting in Eq. (3), Wang et al. (2023b) define geometric perturbation using isotropy scaling, which can be written as

$$\begin{bmatrix} \mathbf{x}_j \\ \mathbf{y}_j \end{bmatrix} = \begin{bmatrix} \theta_s \cdot \cos\theta_r & -\theta_s \cdot \sin\theta_r & \theta_t^{hor} \\ \theta_s \cdot \sin\theta_r & \theta_s \cdot \cos\theta_r & \theta_t^{vrt} \end{bmatrix} [\mathbf{x}_i', \mathbf{y}_i', 1]^\top. \quad (17)$$

Here, they set the rotation to $\theta_r = 20°$, the horizontal and vertical translations to $\theta^{hor} = \theta^{vrt} = 22.4$ pixels, and the scaling factor to $\theta_s = 0.1$.

Among existing DIRECT-based methods, we consider two key baselines: the classic DIRECT-$L_\infty$ algorithm, which employs the $L_\infty$ norm to measure the size of partitions (Wang et al., 2023b), and SimpleDIRECT, which was introduced to improve scalability in high-dimensional applications (Wang et al., 2025)[1]. In contrast, our proposed DIRECT-LSR operates under the $L_2$ norm to ensure stronger theoretical convergence guarantees. For fair comparison, all methods are executed with a maximum partition depth of $H = 6$ and a total query budget of 3000, and we targeted five models, including: ResNet50, WideResNet50, Vit$_{16\times16}$, Large Beit$_{16\times16}$, and Swin Transformer.

---

[1]Wang, Fu, et al. "A Black-Box Evaluation Framework for Semantic Robustness in Bird's Eye View Detection." AAAI. 2025.

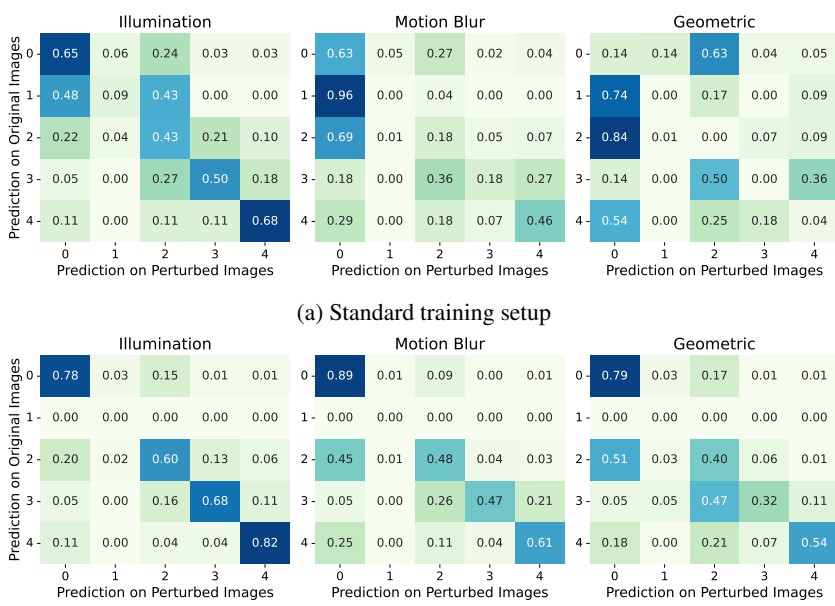

(a) Standard training setup

(b) Semantic augmentation training setup

Figure 7: Averaged prediction transition matrices on the EYEPACS (DR grading) dataset comparing (a) Standard and (b) Augmented training strategies. The matrices show how models' predictions change under illumination, motion blur, and geometric perturbations. The labels 0 to 4 correspond to no, mild, moderate, severe, and proliferative DR, respectively. All values are row normalised.

As reported in Table 5, we compare the attack success rate (ASR), verified accuracy, and the runtime. As shown, the large $\text{BEiT}_{16\times16}$ model demonstrates the strongest robustness overall, consistent with findings in Wang et al. (2023b). DIRECT-LSR achieves the highest attack success rate (ASR) across all five classifiers, while maintaining a reasonable runtime overhead. The additional runtime is attributed to the least squares regression process with a computational complexity of $\mathcal{O}(Hn^3)$. Although both DIRECT and SimpleDIRECT report higher verified accuracy, their lower bounds are based on recorded slopes (Eq. (9)), which lack formal guarantees and may overestimate robustness.

### E.2 POST ANALYSIS

In Fig. 7, 8, and 9, we show the averaged prediction transition caused by different semantic perturbations across all models on EYEPACS, AREDS, and GLC datasets.

On the EYEPACS dataset, semantic perturbations often cause the standard models' predictions to shift toward adjacent severity levels. Although data augmentation substantially improves robustness, the augmented models still struggle to correctly identify mild DR, suggesting a potential trade-off between robustness and fine-grained discrimination. We plan to investigate the underlying reasons for this behaviour in future work.

On the AREDS and GLC datasets, the models are trained to make binary predictions. We can see from Fig. 8 and 9 that the standard trained models are easily misled into classifying advanced AMD as non-advanced and positive GLC as negative, which are Type II errors that are particularly concerning in clinical practice. When trained with data augmentation, the models' robustness has been notably improved, and the Type II error rate has reduced by more than 60%. This improvement is also consistent with the overall performance gains from the data augmentation on these two datasets reported in Table 4.

### E.3 MIXED SEMANTIC PERTURBATION

In this experiment, we evaluate the models' robustness under mixed semantic perturbations, combining geometric transformations ($\gamma = 0.2$), illumination distortions ($\gamma = 0.1$), and motion blur

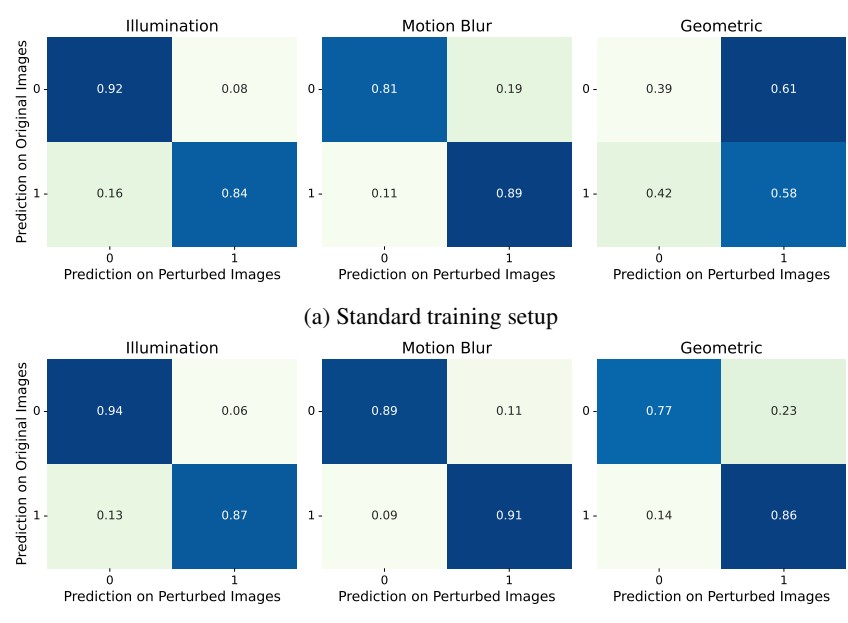

(a) Standard training setup

(b) Semantic augmentation training setup

Figure 8: Averaged prediction transition matrices on the AREDS (AMD diagnosis) dataset comparing (a) Standard and (b) Augmented training strategies. The matrices show how models' predictions change under illumination, motion blur, and geometric perturbations. The labels 0 and 1 correspond to non-advanced and advanced AMD, respectively. All values are row normalised.

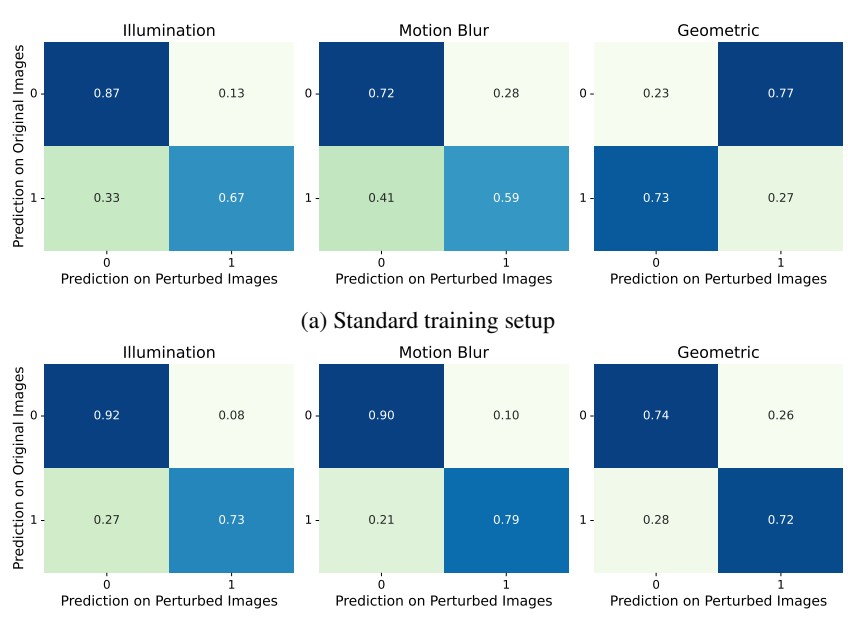

(a) Standard training setup

(b) Semantic augmentation training setup

Figure 9: Averaged prediction transition matrices on the GLC (glaucoma diagnosis) dataset comparing (a) Standard and (b) Augmented training strategies. The matrices show how models' predictions change under illumination, motion blur, and geometric perturbations. The labels 0 and 1 correspond to negative and positive outcomes, respectively. All values are row normalised.

($\gamma = 5$). As shown in Table 6, taking ResNet50 and RetFound for example, even the enhanced models remain highly vulnerable to the combined semantic perturbations, with none achieving validated

Table 6: Model accuracy and verified accuracy under mixed semantic perturbations

| Model | EYEQ | | EYEPACS | | AREDS | | GLC | |
|---|---|---|---|---|---|---|---|---|
| | Acc. | V. Acc. | Acc. | V. Acc. | Acc. | V. Acc. | Acc. | V. Acc. |
| RetFound | 0.152 | 0.0 | 0.246 | 0.0 | 0.234 | 0.0 | 0.04 | 0.0 |
| +Aug. | 0.186 | 0.0 | **0.502** | 0.0 | 0.558 | 0.0 | 0.288 | 0.0 |
| ResNet50 | 0.146 | 0.0 | 0.0 | 0.0 | 0.242 | 0.0 | 0.212 | 0.0 |
| +Aug. | **0.188** | 0.0 | 0.158 | 0.0 | **0.686** | 0.0 | **0.324** | 0.0 |

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

robustness on any of the perturbed examples. This result exposes the limitations of data augmentation in defending against complex, multi-faceted semantic perturbations. It highlights the need for more effective defence mechanisms and for robustness evaluation frameworks that better align with human perception and clinical relevance.

### E.4 FULL BENCHMARK

In Table 7, we report the full benchmark that includes ResNet50 (He et al., 2016), RegNet (Radosavovic et al., 2020), EfficientNet (Tan & Le, 2019), and DenseNet (Huang et al., 2017), and a pure Transformer model (RetFound (Zhou et al., 2023)). , and a hybrid convolution-Transformer model (CoAtNet (Dai et al., 2021)). RegNet and CoAtNet are not listed in Table 2 because they do not achieve the best performance under any of the evaluated settings.