# OpenReview forum: "Semantic Robustness of Deep Neural Networks in Ophthalmology: A Case Study with Colour Fundus Imaging"
_ICLR.cc/2026/Conference — Submitted to ICLR 2026_

### Official Review · Reviewer_wWbD · 2025-10-29

**Soundness:** 3
**Presentation:** 2
**Contribution:** 2
**Rating:** 2
**Confidence:** 4

**Summary:**

This paper investigates the robustness of retinal imaging models to "semantic" corruptions such as geometric distortions, illumination changes, and motion blur (in contrast to adversarial corruptions). These corruptions are common in datasets of retinal images due to practical issues in data collection.
The authors parameterize the corruptions and present an algorithm (DIRECT-LSR) which can find distortion parameter values that are provably close to optimal, in the sense that corruptions will maximally affect classifier performance. Indeed, the found corruptions are shown to degrade performance stronger than corruptions found by PGD-like gradient-based methods. The authors then show that augmenting the training data with such corruptions improves model robustness with respect to these semantic corruptions.

**Strengths:**

- The investigated problem is practically relevant, for the applied field of retinal imaging.
- The parametrization of the corruptions seems reasonable and is explained clearly.
- The proposed algorithm, DIRECT-LSR, could be useful in other contexts as well.
- The investigation covers multiple models and datasets.

**Weaknesses:**

- The scope of the paper is very narrow. I see the potential value of this analysis to the medical field, specifically the subfield working on retina imaging, but for ICLR, this somewhat lacks generality. One would have to pitch the DIRECT-LSR algorithm as a general method, and I could see myself accepting such a paper, but this is not what the paper does.
- The presentation of the paper should be improved:
	- Figure 3 is a bit weird because the lines all overlap, so most of the colors never even show up - probably, a logarithmic y-axis would have been advisable.
	- Figure 4 is poorly designed, I'd recommend to focus on one dataset, then put the rest in the appendix as separate figures.
	- Table 2 is hard to read, it includes a few irrelevant models (as in, models that never perform best), and no confidence intervals or any notion of uncertainty. I would make a selection of models, then have the full table in the appendix. Likewise, I would remove the $L_{min} > 0$ rows and present them in an extra appendix table.
- The related work section is not very convincing, I would have liked to see works more closely connected to robustness of retinal imaging models rather than e.g. an arbitrary autonomous driving paper.
- It seems like train / test splits were not conducted properly. If this is indeed the case, it would be a weakness.
- The four datasets are not sufficiently characterized. How large are they? What is the state-of-the-art classification performance on the datasets?
- The evaluation of human experts was not done perfectly. Ideally, human experts should have been asked to perform a classification task "blindly", in a true nAFC fashion, i.e. without access to the true labels, rather than just stating whether they think crucial features for the true class are visible or not. This would put model performance in perspective much better.
- The limits of all corruption parameters are neither justified nor visualized, making it hard to assess whether they are reasonable (e.g. there are levels of motion blur that would be absurd to expect in retina images).
- The analysis is a bit thin, I would have liked to see e.g. whether the different optimization procedures (DIRECT-LSR, Bayes, random search) find very different optima, whether your attacks lead models to favor a certain class, etc.
- I realize this is a somewhat unfair criticism, but I find it hard to believe that random search outperforms PGD-style gradient-based optimization for the task of finding optimal corruptions. I must imagine that this is indicative of sub-par implementation or poor hyperparameters. Can the authors comment on this? (Maybe the random search effectively covers the entire parameter space?)
- Training the models in table 2 with exactly the same setup does not seem very principled to me. Different training setups might be optimal for different models, so I would find it more convincing if all models were trained in a way that yields maximum performance (up to the limit of the respective architecture and model size). Another reasonable decision would have been to compute-match models. I would have also liked to see reference performance values on these tasks from literature.
- Again a somewhat unfair point, but the paper lacks certain shibboleths that one would expect from groups that work in this field. For example, the phrase "out-of-distribution robustness" does not appear even once, although this is exactly what this paper is about at its core. I have tried to down-weight this point in my assessment, but it does not instill confidence.

Overall, I feel conflicted about this paper. In principle, I see the value of the idea of finding provably difficult points in bounded parameter spaces for parametric corruptions, and I can imagine that evaluating robustness to such corruptions is relevant in the retinal imaging field. The biggest argument in favor of the paper is that the proposed algorithm could be useful in other settings. But the paper itself suffers from poor presentation, and there are some "smells" (such as the train-test-split issue, model comparison on equal footing, bad PGD performance, etc). The paper also has a very narrow topic scope and many questions remain open. Based on the related work, I am also lacking perspective on e.g. what models are considered gold standard for these tasks, and how robustness has been evaluated in the literature. I'm submitting an initial rating of 2 for now, but I am curious about other reviewers' thoughts and generally willing to increase this score, provided that presentation is improved and the paper is championed by someone else.

**Questions:**

- I would find the paper more valuable if the idea was to implement a benchmark, where people submit retinal imaging models, and maximally strong "semantic adversarials" are created specifically for these models, eventually leading to a public leaderboard of the most robust retinal imaging models. Do you plan on implementing this?
- What was the train- / test-split of the datasets? It reads in line 440 as if the dataset was not split, and validation was done on a random subset of the training set.
- What is the Schwefel function? This should probably explained in at least one sentence.

---

> ### Author Response · Authors · 2025-11-20
>
> Thank you for the thorough assessment of our work and valuable comments. We provide a point-by-point response as follows:
> 1. We thank the reviewer for recognising the potential general value of DIRECT-LSR. While the main focus of the paper is on colour fundus imaging, where semantic robustness is clinically important and underexplored, the optimisation method itself is not domain-specific. To illustrate this, we additionally report results on ImageNet classifiers (Appendix), which show that DIRECT-LSR performs consistently on natural images as well. This supports the general applicability of our approach beyond the medical setting.
> 2. - For Figure 3, the curves overlap because DIRECT-LSR is a deterministic optimiser and the minimum values obtained under different maximum partition depths are naturally very similar.
>    - For Figure 4, we agree that the current layout is dense. Our intention was to present, within a single figure, both the outcome of the different optimisation strategies and the corresponding expert assessments. This required a tall and compact layout under the page-limit constraints. Since each dataset represents a different clinical task in colour fundus imaging, we believe it is important to show all results in the main paper to provide a complete picture of the models' robustness. Nevertheless, we can revise the figure layout to improve readability.
>    - For Table 2, thank you for the suggestion. We have now moved the two models that never achieve the best performance to the appendix. We prefer to keep the $L_{\min}>0$ row in the main table as lower bound estimation is a core feature of DIRECT-LSR and provides a stricter robustness validation.
> 3. In the related work section, we highlighted the previous studies most relevant to ours on the methodological side. To the best of our knowledge,  formal evaluations of models’ semantic robustness are very limited in the domain of colour fundus imaging. We have updated the related work section to highlight the research gap.
> 4. We confirm that the train/test splits were conducted properly. The dataset sizes and the exact train–test partitions are provided in Appendix Table 3, and the corresponding clean-performance metrics (accuracy and F1) are reported in Appendix Table 4. These results verify that all models were evaluated on held-out test sets following standard practice.
> 5. We appreciate the reviewer’s suggestion. Our goal in the human assessment is to confirm the clinical plausibility of the semantic perturbations. For this purpose, we asked experts to evaluate whether the perturbations alter class-relevant features. We provided the ground-truth label because conducting a full blind diagnostic task would require substantially more annotation effort and would also introduce additional variability from human diagnostic errors and inter-reader biases [Krause et al.]. Our design therefore isolates the effect of the perturbation itself.
> 6. The corruption ranges were chosen to reflect plausible acquisition variations, and their reasonableness is validated through the human expert assessments. Our goal is to identify the _maximum_ perturbation level that can degrade model performance without affecting human interpretability, which is hard to measure in an algorithmic way. The expert evaluations therefore serve as an empirical justification that the perturbation limits remain clinically realistic and do not produce absurd or implausible artefacts.
> 7. We thank the reviewer for this suggestion. We have added new visualisations and analyses showing how the models’ predictions change under different perturbation types. These analyses are based on the results in Table 2 rather than Figure 4. As DIRECT-LSR clearly finds stronger perturbations under the same setup compared with the baselines, we consider it more informative, within the scope of this work, to focus on analysing the models’ behaviour across different tasks.
> 8. This phenomenon has been reported before: Engstrom et al. showed that PGD-style gradient attacks can underperform random search for geometric perturbations. Our results extend this observation to illumination and motion-blur settings. This reflects that the loss landscape over semantic parameters can be highly non-concave, making gradients uninformative, rather than indicating an implementation or hyperparameter issue.

---

> > ### Author Response · Authors · 2025-11-20
> >
> > 9. We agree that different architectures may achieve a higher clean accuracy under different training setups. However, varying the training setups across models would introduce additional experimental factors, as training choices can also affect robustness. For example, higher clean performance obtained through task-specific tuning may come at the cost of reduced robustness due to overfitting. To ensure a fair and controlled comparison, we therefore trained all models under the same setup and evaluated their best checkpoints. This avoids mixing robustness differences with architecture-specific training settings.
> > 10. We thank the reviewer for the comment. Our work focuses on semantic perturbations that naturally occur during image acquisition and should not change the underlying class semantics. For this reason, our setting should be viewed as "in-distribution robustness", rather than "out-of-distribution robustness". This is also reflected in Figure 4, where the valid perturbations should reduce model performance while leaving human expert assessments unaffected.
> >
> > Q1: We will release the code on GitHub and consider hosting a benchmark for public submission.
> >
> > Q2: The dataset sizes and the train–test partitions are as follows, we have also reported the same table in the Appendix Table 3.
> > | Task                                  | Train Size | Test Size |
> > |---------------------------------------|------------|-----------|
> > | Image quality check (EYEQ)            | 12,545     | 16,251    |
> > | Diabetic retinopathy grading (EYEPACS)| 35,125     | 53,574    |
> > | Age-related macular degeneration (AREDS) | 8,420   | 8,613     |
> > | Glaucoma (GLC)                        | 4,822      | 4,822     |
> >
> > Q3: The Schwefel function is a challenging test function for global optimisation algorithms due to its large number of local minima. It can be written as $f(x)=418.9829d-\sum_{i=1}^{d} x_i \sin(\sqrt{\|x_i\|)}$, where $d$ is dimensionality and $x\in[-500,500]$. We have added the corresponding description in the appendix for clarity.
> >
> > Engstrom et al. Exploring the landscape of spatial robustness. In ICML, 2019.
> >
> > Krause et al. Grader variability and the importance of reference standards for evaluating machine learning models for diabetic retinopathy. Ophthalmology 125.8 (2018): 1264-1272.

---

### Official Review · Reviewer_NijQ · 2025-10-31

**Soundness:** 2
**Presentation:** 2
**Contribution:** 2
**Rating:** 6
**Confidence:** 3

**Summary:**

This paper presents a thorough and well-motivated framework for evaluating the semantic robustness of deep neural networks (DNNs) in retinal image diagnosis. Rather than relying solely on pixel-level adversarial perturbations, the authors propose a semantic perturbation framework that operates at the level of medically meaningful transformations (e.g., illumination, lesion intensity, vessel sharpness). The goal is to assess whether retinal disease classifiers remain stable under clinically relevant but semantically preserving changes. This is a timely and practically significant contribution to the intersection of adversarial robustness, explainable AI, and medical imaging, with strong potential implications for clinical reliability and model validation.

**Strengths:**

1. The work addresses a real and pressing gap: most medical AI robustness studies focus on pixel-level noise or domain shifts, ignoring semantic perturbations aligned with clinical reasoning. The proposed semantic perturbations are interpretable, controllable, and grounded in medical semantics, which enhances both transparency and clinical adoption potential.
2. The framework systematically defines semantic dimensions and corresponding transformation operators. The perturbations are implemented in a way that maintains plausible clinical realism, avoiding synthetic artifacts.
3. Evaluation across multiple datasets and models demonstrates the framework’s generality. The correlation analysis between semantic robustness and adversarial robustness is particularly insightful—it shows they are distinct but complementary.

**Weaknesses:**

1. The framework is empirical and descriptive; it lacks formal definitions or theoretical guarantees of semantic robustness (e.g., invariance under a semantic transformation group). A more formal link to robustness theory (e.g., Lipschitz continuity under semantic metrics) would strengthen its academic rigor.
2. The chosen perturbations, while clinically plausible, are manually curated and limited to a few dimensions. There is no discussion on how to generalize or learn semantic perturbations automatically (e.g., via generative models or disentangled representations).
3. Experiments focus on binary diabetic retinopathy grading. It remains unclear how well the framework generalizes to multi-class or multi-label medical tasks (e.g., glaucoma, AMD). The lack of external validation on unseen imaging modalities (e.g., OCT) limits generalizability.

**Questions:**

1. How do you define the boundary between semantic and non-semantic perturbations, especially when pixel-level changes may indirectly alter semantic meaning?
2. Could your semantic perturbation framework be adapted for unsupervised discovery of semantic factors using disentangled or generative representations?
3. Do you have any insights into why ViT architectures (if included) appear more semantically stable than CNNs, or vice versa?

---

> ### Author Response · Authors · 2025-11-20
>
> Thank you for the positive comments and insightful questions. We provide our point-by-point responses below:
>
> **W1:** We appreciate the reviewer’s suggestion. Indeed, having theoretical guarantees for semantic robustness would be ideal. Unfortunately, while our method can provide a lower-bound estimation, it cannot guarantee convergence under a limited query budget, so we cannot claim formal verification at this point. We position this work as an empirical assessment of model robustness on colour fundus imaging and formulate the semantic perturbations in a quantitative and optimisable way, which allows us to control the perturbation strength and makes them suitable for optimisation with DIRECT-LSR. The approach is compatible with future theoretical developments, and any semantic perturbation that can be formally defined in the future could be incorporated into the DIRECT-LSR framework.
>
> **W2:** Learning semantic perturbations automatically is indeed an interesting direction. In this work, we focus on low-level perturbations that have clear and optimisable parameterisations, which allow us to control the perturbation strength and apply our optimisation framework in a reliable way.
>
> For higher-level perturbations introduced by methods such as generative models or disentangled representations, it is currently difficult to use them for formal robustness evaluation because there are no quantitative bounds on how much semantic change is introduced. Without such bounds, it becomes challenging to ensure consistent or interpretable robustness assessment. Deriving such bounds for generative semantic perturbations would be a valuable research direction, but it falls outside the scope of our work at this moment.
>
> **W3:** In our experiments, we have included four tasks: a 5-class diabetic retinopathy grading task, a 3-class colour fundus image quality assessment task, and two binary diagnosis tasks for glaucoma and AMD. These cover both multi-class and binary settings.
> In this work, we focus on colour fundus imaging and did not conduct experiments on OCT. To show generalisability, we include additional experiments on the ImageNet dataset (Appendix, Table 5). Because the proposed method is query-based, it could be applied to existing OCT models and multi-label models, but this would require extra work to define clinically meaningful perturbations for OCT images and to formulate the objective function for multi-label settings, respectively.
>
> **Q1:** We use the term _semantic perturbation_ mainly to distinguish our setting from norm-bounded pixel-level perturbations, which is the conventional terminology in robustness research.  Pixel-level perturbations are commonly restricted by a small norm ball, which should not cause semantic changes. However, if a perturbation does alter the semantic content, that means the ground-truth label might change accordingly. In this case, we believe it is necessary to include human assessment to confirm that the perturbed images are still interpreted consistently by human experts, and to compare this with the model outputs. This helps verify whether the perturbations affect the model’s predictions without altering human judgment.
>
> **Q2:** Potentially yes. Our method could be applied in a disentangled or generative representation space to produce higher-level semantic perturbations. However, as mentioned in our earlier response to W2, the main challenge is defining a meaningful perturbation space that reflects clinically relevant and realistic changes.
>
> **Q3:** Thank you for this insightful question. Our observations on this point are mixed. In retinal imaging, prior work and our own experience show that CNNs often achieve better performance on clean images. One possible reason is that most retinal datasets are relatively small, which is usually sufficient for training CNNs but may not be enough for ViTs. This could affect robustness, as an underfitted model may not exhibit stable generalisation.
>
> On the other hand, because ViTs process images using global attention, they may be more stable against certain types of semantic perturbations. In our experiments, RetFound, a ViT-based backbone pretrained on 1.6 million retinal images, showed comparatively considerable robustness to geometric perturbations.

---

### Official Review · Reviewer_WJMB · 2025-11-02

**Soundness:** 1
**Presentation:** 1
**Contribution:** 1
**Rating:** 0
**Confidence:** 5

**Summary:**

This paper proposes a framework for evaluating semantic robustness of deep neural networks in color fundus photography. The authors focus on three types of distortions, namely geometric transformation, illumination changes, and motion blur. The paper tries to utilize a non-gradient optimization method based on the DIRECT algorithm.

This paper should be rejected because there are significant shortcomings including the following points:

1- The proposed algorithm (DIRECT-LSR) is a simplistic modification to the original DIRECT algorithm.

2- The paper does not provide any justification in using the DIRECT-LSR to evaluate robustness. In fact, the authors do not formally define what robustness means in the context used in this paper.

3- The paper claims be able to evaluate robustness of neural networks to semantic perturbations and changes. However, the experiments simply use geometric transformations, illumination changes, and motion blur as examples of semantic perturbations. These are not in fact changing the semantic context of the images, and thus are practically irrelevant. Again, this is due to the fact that the paper doesn't clearly and formally define robustness and semantic sensitivity of neural networks.

4- The paper does not provide any comparison with other state-of-the-art algorithms designed for assessing robustness.

5- Evaluation against human expert assessments and protocols used for this evaluation is not clearly discussed.

6- This paper is an application paper, focused on a narrow domain (retina fundus images). Its applicability beyond this domain is questionable.

**Strengths:**

The paper attempts to evaluate the robustness of neural networks to semantic perturbations.

**Weaknesses:**

The paper suffers from many weaknesses with major shortcomings listed below:
1) Lack of novelty. The paper simply applies a least-squar-regression to the DIRECT algorithm.
2) Lack of motivation behind the use of this optimization algorithm. Why not evaluate with a gradient-based approach?
3) Lack of comparative evaluations against the state-of-the-art.
4) Lack domain generalizability. It is unclear how this method can be applied beyond fundus images.
5) Insufficient experimentation. The paper simply uses three types of image manipulation. Would this method work for adversarial attaches? Or actual semantic perturbations, like content manipulation? The paper also only evaluates six neural nets for robustness. The claim that the proposed method can evaluate neural network robustness should design a more comprehensive framework for evaluating major neural networks' robustness.

**Questions:**

None.

---

> ### Author Response · Authors · 2025-11-20
>
> We thank the reviewer's effort on the comments. Our responses are provided below.
> 1. We respectfully disagree with the assessment that our work lacks novelty. Least-squares-regression Lipschitz constant estimation has only been introduced recently in [a], and its integration into the DIRECT framework has not been previously explored. Our work is the first, to our knowledge, to incorporate this estimation technique into DIRECT in a principled manner.  This integration enables the optimiser to maintain a more accurate and adaptive estimate of the lower bound of the objective function compared with the existing approach in [b]. This directly addresses a key limitation of applying heuristic global optimisation in robustness validation. Therefore, our methodological contribution is not a simple application of an existing technique, but a substantive extension that improves the practical effectiveness of DIRECT in robustness evaluation scenarios.
> 2. The main motivation for adopting DIRECT is that existing methods do not efficiently evaluate semantic robustness. As summarised in our related work, most empirical studies rely on randomised search, which provides weak guarantees and tends to overestimate robustness. White-box verification methods, designed for norm-bounded pixel perturbations, struggle to handle the large, spatially non-uniform pixel distortions induced by semantic transformations. Similarly, randomised smoothing cannot obtain the optimal worst-case perturbation configuration. In contrast, DIRECT, a global derivative-free optimiser, systematically explores the semantic parameter space and yields stronger and more reproducible robustness evaluations.
> **Our paper already includes a first-order gradient-based baseline** (Figure 4). As reported, the gradient-based approach **does not outperform DIRECT and can even underperform random search**. This aligns with an existing work [c] for geometric perturbations, and our results extend the observation to illumination and motion-blur settings.
> 3. Semantic robustness remains relatively underexplored, and only a limited number of evaluation strategies have been adopted in prior work. In our experiments, we therefore compared DIRECT against the most representative and widely used baselines, including random search, Bayesian optimisation, and a first-order gradient-based attack. These baselines should cover the dominant optimisation paradigms previously applied to semantic robustness evaluation. If the reviewer has specific baselines in mind, we would appreciate the pointers and are happy to include them if feasible.
> 4. Our method is not restricted to fundus images. To demonstrate domain generalisability, in the appendix, we have already provided results evaluating semantic robustness on 5 ImageNet classifiers. We also compared DIRECT-LSR against recent variants of DIRECT on the ImageNet dataset. The results show that DIRECT-LSR consistently performs competitively across these non-medical, large-scale natural image models, indicating that the proposed approach is not domain-specific and generalises beyond fundus imaging.
> 5. We would like to clarify the scope of this paper. Our goal is to develop an efficient optimisation strategy for evaluating semantic robustness, rather than to introduce new forms of semantic perturbations or to exhaustively benchmark every possible model architecture. We therefore focus on the three types of image manipulations that (i) represent clinically meaningful semantic variations in colour fundus imaging, and (ii) possess well-defined formulations suitable for optimisation. These choices enable a controlled and rigorous evaluation of the models' semantic robustness, which is the core contribution of this work.
>    With respect to model coverage, we have evaluated six representative and domain-relevant neural networks, covering the major families currently used in fundus imaging. While it is impractical to include all existing architectures, our results demonstrate vulnerability across diverse backbones. The additional ImageNet experiments (in the Appendix) further indicate that the approach generalises beyond fundus imaging.
>    Regarding broader semantic perturbations, such as adversarial content manipulation, many of these transformations either lack a mathematically tractable formulation. Our method is orthogonal to the formulation of new semantic transformations, and could in principle be applied whenever the semantic perturbation can be explicitly specified. If the reviewer has specific neural networks or well-defined semantic transformations that are relevant to this study, we would be grateful for the suggestions and are happy to consider incorporating them.
>
> [a] Huang et al. On the sample complexity of Lipschitz constant estimation. ICLR, 2024.
>
> [b] Wang et al. Towards verifying the geometric robustness of large-scale neural networks. AAAI, 2023.
>
> [c] Engstrom et al. Exploring the landscape of spatial robustness. ICML, 2019.

---

### Meta-Review · Area_Chair_MrYA · 2025-12-08

**Summary:**

This submission presents a case study on the semantic robustness of DNN models for colour fundus imaging. The reviewers are concerned about the unclear novelty of the proposed DIRECT-LSR method, insufficient theoretical grounding (particularly the lack of a formal definition of semantic robustness), and the limited comparative evaluation against existing robustness approaches. Reviewers also found the experimental scope narrow, with questions about dataset handling, choice and validation of semantic perturbations, and limited generalizability beyond ophthalmic imaging. Presentation issues further reduce the clarity of the paper (unclear figures, missing baseline context, and incomplete positioning of the work relative to robustness literature).

**Reviewer Concerns:**

The authors provided clarification on the novelty of their method by explaining how least-squares Lipschitz estimation has not previously been integrated into the DIRECT framework, and they justified the choice of their optimisation approach relative to gradient-based methods, citing prior work demonstrating similar limitations. They also addressed questions around generalizability by pointing to ImageNet experiments included in the appendix, and clarified dataset splits and evaluation protocols in response to concerns about experimental validity.

However, several concerns remain unresolved. The most significant is the lack of a formal definition or theoretical framing of semantic robustness, which multiple reviewers highlighted as a missing foundational component. The rebuttal acknowledges this gap rather than resolving it. Additionally, the issue of scope and positioning persists: although supplementary ImageNet experiments help, the paper is still primarily framed as a domain-specific medical application rather than a general robustness framework, and hence is too narrow in scope for ICLR.

**Reviewer Scores:**

Reviewer WJMB (score 0) expressed the most severe concerns, citing lack of novelty, unclear motivation, weak experimental design, missing comparisons, methodological gaps, and lack of clarity. While the rebuttal addresses some points—particularly justification for DIRECT-LSR—it does not fully resolve the missing theoretical definition, unclear positioning w.r.t. existing literature and limited scope. It is likely that the reviewer would retain the score 0.

Reviewer NijQ (score 6) was positive, identifying the paper as valuable and timely, while noting limitations around theory, scope, and generalization (criticism that is shared with both other reviewers). I believe that the reviewer would lower the score to 4 based on the rebuttal and the other reviews.

Reviewer wWbD (score 2) provided a mixed rating, being intrigued by the idea, but concerned about the narrow scope of the work, which was not addressed sufficiently in the rebuttal. I believe that the reviewer would retain the score 2.

---

### Decision · Program_Chairs · 2026-01-26

Reject